# United States Pharmacopeia (USP) Safety Review of *Gamma*-Aminobutyric Acid (GABA)

**DOI:** 10.3390/nu13082742

**Published:** 2021-08-10

**Authors:** Hellen A. Oketch-Rabah, Emily F. Madden, Amy L. Roe, Joseph M. Betz

**Affiliations:** 1U.S. Pharmacopeial Convention, 12601 Twinbrook Parkway, Rockville, MD 20852, USA; emily.madden@usp.org; 2The Procter & Gamble Company, Cincinnati, OH 45202, USA; roe.al@pg.com; 3Office of Dietary Supplements, National Institutes of Health, Bethesda, MD 20892, USA; betzj@od.nih.gov

**Keywords:** *γ*-amino butyric acid, *gamma*-amino butyric acid, GABA, safety review, adverse effects, adverse events, dietary supplements, interactions, hypotension, 4-aminobutanoic acid

## Abstract

*Gamma*-amino butyric acid (GABA) is marketed in the U.S. as a dietary supplement. USP conducted a comprehensive safety evaluation of GABA by assessing clinical studies, adverse event information, and toxicology data. Clinical studies investigated the effect of pure GABA as a dietary supplement or as a natural constituent of fermented milk or soy matrices. Data showed no serious adverse events associated with GABA at intakes up to 18 g/d for 4 days and in longer studies at intakes of 120 mg/d for 12 weeks. Some studies showed that GABA was associated with a transient and moderate drop in blood pressure (<10% change). No studies were available on effects of GABA during pregnancy and lactation, and no case reports or spontaneous adverse events associated with GABA were found. Chronic administration of GABA to rats and dogs at doses up to 1 g/kg/day showed no signs of toxicity. Because some studies showed that GABA was associated with decreases in blood pressure, it is conceivable that concurrent use of GABA with anti-hypertensive medications could increase risk of hypotension. Caution is advised for pregnant and lactating women since GABA can affect neurotransmitters and the endocrine system, i.e., increases in growth hormone and prolactin levels.

## 1. Introduction

*Gamma*-amino butyric acid (GABA) is a four-carbon, non-protein amino acid that is widely distributed throughout biological organisms, including plants, animals, and microorganisms. The compound GABA was first synthesized in 1883 and, at that time, was thought to be a metabolite in plants and microbes only. Subsequent research showed that GABA is the chief inhibitory neurotransmitter in the mammalian central nervous system [1,2]. GABA is also a neurotransmitter in a range of invertebrate phyla, including arthropods, echinoderms, and platyhelminths. The functional properties of GABAergic neurons in simple gastropod systems may provide insight into the role of this neurotransmitter phenotype in more complex brains [3]. Two other aminobutyric acid (ABA) isomers exist, α-aminobutyric acid (AABA), also known as homoalanine, and β-aminobutyric acid (BABA). Increased levels of AABA have been linked to some diseases like tuberculosis and pediatric metabolic diseases, such as Reye’s syndrome. Elevated AABA levels also have been investigated as a possible biomarker of alcohol-induced liver injury, sepsis, malnutrition, and multiple organ failure [4,5,6]. BABA is a natural plant product that has been shown to increase plant resistance to diseases and, when applied to plants, can increase their resistance to abiotic stress [7]. The current review is about GABA only.

Natural GABA was first isolated from potato tuber tissue [8]. In plants and microbes, GABA is an integral part of the Krebs cycle and has been observed to increase rapidly during environmental stresses, indicating a potential role in stress response [9]. In animals, GABA functions as the major inhibitory neurotransmitter in the central nervous system, and in rats, it is estimated that at least one-third of all central nervous system neurons utilize GABA as their primary neurotransmitter [10]. Thirty percent of human cerebral neurons contain GABA, which affects almost all neuronal activities [11,12]. In biological systems, GABA is synthesized from glutamate via the GABA shunt [10]. The synthesis is catalyzed by the enzyme L-glutamic acid decarboxylase (GAD), with the help of pyridoxal phosphate (P5P, the active form of vitamin B6) as the co-factor. GABA is metabolized by gamma-aminobutyrate transaminase into an intermediate metabolite, succinate semi-aldehyde, which is then reduced to gamma-hydroxybutyrate or oxidized to succinate and eventually converted to CO_2_ and water via the citric acid cycle [10].

GABA has been investigated for its effects on reducing stress and enhancing sleep in human studies [13] and for its other biological activities, which include anti-hypertension, anti-diabetes, anti-cancer, antioxidant, anti-inflammation, anti-microbial, and anti-allergy effects [14]. 

In the U.S., GABA is marketed as an ingredient in a number of dietary supplements (DS). Some of the purported uses of GABA include to relieve anxiety, elevate mood, relieve premenstrual syndrome (PMS), increase lean muscle mass, burn fat, stabilize blood pressure, and relieve pain [15,16]. GABA is a popular ingredient in sports dietary supplements and other wellness dietary supplements. Some forecasting companies estimate that the global market size for GABA dietary supplements will increase significantly from USD 38 million in 2019 to USD 50 million by the end of 2026 [17]. 

Because of the extensive use of GABA as an ingredient in dietary supplements in the U.S. market, USP decided to develop a dietary supplement quality monograph. Prior to developing a dietary supplement monograph, USP conducts a dietary supplement admission evaluation that includes a safety evaluation intended to determine whether an ingredient is associated with any serious risk(s) to health that would preclude its admission for monograph development. This evaluation of GABA was conducted according to USP guidelines for the admission of dietary ingredients into the monograph development process [18] and includes an assessment to determine whether GABA presents a serious risk to human health. This comprehensive review examined the use of GABA to control blood pressure and as an ergogenic in sports supplements; USP also examined its potential interaction with antihypertensive medications.

## 2. Literature Search Method

A search was done in PubMed using the terms GABA, gamma-aminobutyric acid, and aminobutanoic acid, combined with the following terms: safety, clinical trials, case reports, reviews, humans, in vitro, adverse effects or side effects, pharmacokinetics, and phytochemistry, covering up to March 2021. Figure 1 is a graphical representation of the search strategy, showing articles retrieved from PubMed and the number of articles that are included in this review. The following electronic databases were also searched: Google Scholar, TOXLINE, ScienceDirect, REPROTOX, and The Cochrane Library (from inception to March 2021). Additional publications were identified by searching the reference lists of relevant journal publications, review articles, and books.

## 3. GABA Chemistry, Natural Sources, and Metabolism

GABA is a four-carbon non-protein amino acid, a butanoic acid with an amino substituent located at C-4 position. The molecular formula for GABA is C_4_H_9_NO_2_. The IUPAC name is 4-aminobutanoic acid (Figure 2), the CAS number is 56-12-2, and the UNII code number is 2ACZ6IPC6I [19]. The molecular weight is 103.12 g/mol, and the melting point is 202 °C. GABA is also referred to as 4-aminobutanoic acid, piperidic acid, and piperidinic acid. Other identifiers for GABA can be found in PubChem [20]. GABA is a crystalline substance, white to light yellow in color, and is freely soluble in water but insoluble or poorly soluble in other solvents.

### 3.1. Natural Sources of GABA

GABA is ubiquitous in plants, where it is primarily synthesized from glutamic acid via the glutamate carboxylase enzyme [21]. It has been shown to increase significantly in plants following environmental and other stresses, such as drought, increased salinity, wounding, hypoxia, infection, and germination. A number of edible and medicinal plants contain GABA at different levels. For example, in tomato fruit (*Solanum lycopersicum*), it accumulates as the fruit matures until the ripening stage, when its catabolism is accelerated [22]. A study of GABA content of select uncooked foods found that some contain modest amounts of GABA. Brown rice germ (718 nmol/g), sprouted cereals (300–400 nmol/g), and spinach (414 nmol/g) had the highest levels of GABA [23,24]. Other studies demonstrated similar findings in sprouted or germinated beans, including adzuki beans (*Vigna angularis* (Willd.) Ohwi and H. Ohashi) [25], lupin (*Lupinus angustifolius* L.) [26], and soybean (*Glycine max* L.) [27] when compared to ungerminated beans. Similarly, grains, such as oats (*Avena sativa* L.), wheat (*Triticum aestivum* L.), and barley (*Hordeum vulgare* L.), have been shown to contain GABA [28]. In some fermented foods, GABA occurs at much higher levels. The traditional Korean food kimchi is reported to contain 2667 to 7225 nmol GABA/g [29], while Japanese green tea leaves have been reported to contain 9697–19,395 nmol of GABA/ g on a *w/w* basis [30]. Other lactic acid-fermented foods, such as cured meats and cheeses, also contain high amounts of GABA [31]. Humans have been exposed to GABA in fermented foods since people started consuming such foods.

### 3.2. Commercial Sources of GABA

GABA can be produced via three main methods: (i) microbial fermentation, (ii) enzymatic biocatalysis, and (iii) chemical synthesis. Microbial fermentation is the preferred manufacturing method for commercial use, and although bacteria and fungi are good sources of GABA production, lactic acid bacteria genera have been most extensively used for GABA production. In one manufacturing method, production begins with a fermentation broth containing monosodium glutamate, glutamic acid, yeast extract, glucose, and glycerin fatty acid ester in water. The mixture is sterilized and then inoculated with *Lactobacillus hilgardii* strain K-3 and allowed to ferment for a number of days. Thereafter, the broth is sterilized, undergoes several filtration steps, and then is spray dried to form a powder [11]. A number of patents describing GABA production via fermentation have been filed [32,33,34]. Solid-state fermentation (SSF) is another strategy that has shown promise in the production of GABA. For example, fermentation of *Vicia faba* by *Lactobacillus plantarum* VTT E-133328 resulted in production of 626 mg/kg of GABA [35]. 

Commercial production of GABA via enzymatic biocatalysis is achieved from glutamic acid after decarboxylation in a reaction catalyzed by glutamate decarboxylase (GAD), with the help of the coenzyme pyridoxal-5′-phosphate, to form GABA [11]. However, because of the complexity and associated high cost of purification for GAD as well as constraints in its stability and reusability, this method of production in not viable for commercial application [36,37]. 

Chemical synthesis of GABA can be achieved by a number of routes. In one such method, GABA production involves the carboamination reaction of alkenes, catalyzed by copper complex transfers [38]. Other methods are more complex, including up to five reaction steps that are generally expensive, produce unwanted by-products, and require use of hazardous reagents, making chemical synthesis unfavorable to use commercially [37]. 

### 3.3. Metabolism of GABA

In animals, GABA is synthesized from glutamate via the GABA shunt pathway illustrated in Figure 3 [39]. The synthesis is catalyzed by the enzyme L-glutamic acid decarboxylase (GAD) with the help of pyridoxal phosphate, the active form of vitamin B6, as the co-factor. GABA is metabolized by gamma-aminobutyrate transaminase into an intermediate metabolite, succinate semi-aldehyde, which can then be reduced to gamma-hydroxybutyrate or oxidized to succinate and eventually converted to CO_2_ and water via the citric acid cycle [39].

## 4. Pharmacopeial Quality Standards for GABA

The proposed USP monograph draft currently under development defines GABA to contain not less than (NLT) 98.0% and not more than (NMT) 102.0% of gamma-aminobutyric acid (C_4_H_9_NO_2_), calculated on the dried basis. The monograph proposes identification using infrared (IR) and high-performance liquid chromatography (HPLC) in comparison with the USP Reference Standard for Gamma Aminobutyric Acid. For the assay, HPLC is proposed to determine the constituents and acceptance criterion of 98.0–102.0%. Impurity limits are proposed to be set for chloride and sulfate, with acceptance criteria of NMT 0.05% and 0.03%, respectively, determined following the methods in *USPNF* General Chapter (*USPNF* GC) <221> *Chloride and Sulfate*. Related compounds are limited to 0.5% for individual impurities and NMT 2.0% for total impurities. Other specific tests in the monograph are for pH (acceptance criteria, 6.5–7.5), determined according to *USPNF* GC <791> pH, and loss on drying, according to *USPNF* GC <731> *Loss on Drying* with an acceptance criterion of NMT 0.5%. 

Other surveyed pharmacopeias and health authorities had no monograph for GABA (Japanese Pharmacopoeia, European Pharmacopoeia, International Pharmacopoeia of the World Health Organization, British Pharmacopoeia, Pharmacopoeia of the People’s Republic of China, and monographs in Canada’s Natural and Non-Prescription Health Products Directorate (NNHPD)).

## 5. Regulatory Status of GABA and Intake Levels

In the U.S., GABA as an amino acid meets the definition of a dietary ingredient under 201(ff)(1)(D) and is available in numerous products marketed as dietary supplements and is listed in the United Natural Products Alliance (UNPA) list of dietary ingredients that were in the market prior to the passing of DSHEA in 1994. Although the UNPA list is not officially recognized, the presence of GABA in this list is an indication that GABA may have been used as a dietary supplement prior to the enactment of DSHEA in 1994. As of 14 April 2021, the Dietary Supplement Label Database (DSLD) contained 644 products that had GABA listed anywhere on the label [40]. Two GRAS notices (GRN00257 and GRN00595) were submitted in 2008 and 2015 to the FDA for consideration [41]. Both notices were submitted by the same company and were not reviewed by the FDA as the company withdrew both notices shortly after submission [42]. According to FDA’s *Substances Added to Food* database, GABA (as 4-aminobutyric acid) is used for technical effect as a flavoring agent or adjuvant [43]. According to 40 CFR Part 180-Tolerance and Exemptions for Pesticide Chemical Residues in Food (40 CFR § 180.1188), GABA is exempt from the requirement of a tolerance on all food commodities when used in accordance with good agricultural practice. The Environmental Protection Agency (EPA, Washington, DC, USA) evaluated GABA and decided that it met the statutory requirement of reasonable certainty of no harm [44].

In other countries, GABA is regulated as a medicinal agent or drug. In Canada, GABA is recognized as a medicinal ingredient. A total of 119 active natural health licenses for products containing GABA as a medicinal ingredient were listed as of April 2021 in the Health Canada Licensed Natural Health Products Database (LNHPD) [45]. In Europe, GABA is an ingredient in food supplements. In 2009, the European Food Safety Authority (EFSA) Panel on Dietetic Products, Nutrition, and Allergies provided a scientific opinion on health claims related to GABA and cognitive function and concluded that a cause-and-effect relationship had not been established between intake of GABA and the claimed cognitive functions [46]. In Australia, GABA is not listed on the Australian Register of Therapeutic Goods (ARTG, Woden [ACT], Australia) and is not permitted for use in listed medicines; it is currently not used in any therapeutic goods supplied in Australia. In New Zealand, a document from Medsafe mentions GABA as a Controlled Drug B1 under the Misuse of Drugs Act 1975 [47].

To determine GABA-intake levels in dietary supplements, various databases were searched using the search terms GABA, gamma-aminobutyric acid, and 4-aminobutanoic acid. The manufacturers′ recommended intake on the labeling delivers GABA in the range of 1.5 µg–3000 mg/day, although for the majority of products, recommended intake is 100 mg in divided doses per day. Most products listed in the DSLD carry label statements indicating that products are intended for use by adults 18 years or older. Some products include a label warning stating that: 

*Some individuals may experience a minor tingling of the skin and/or slight shortness of breath shortly after taking GABA. This is characteristic of this amino acid and quickly subsides*.[40]

The NNHPD monograph for Cognitive Function Products recommends a daily intake of 50–3000 mg GABA that does not exceed 750 mg per single dose; it also says to consult a healthcare practitioner for use of products providing 300 mg/day or more when GABA is used for longer than 4 weeks [48]. 

GABA was evaluated by the Joint FAO/WHO Expert Committee on Food Additives (JECFA, Rome, Italy) to determine its safety as a food additive or contaminant. The JECFA committee concluded that the levels of GABA in the body tissues arising from consumption of foods containing GABA as a flavoring agent would be biologically insignificant (in the US population, estimated at 0.1 µg/day) and therefore would present no safety concern [49]. 

## 6. Clinical Evidence of GABA Safety

### 6.1. Clinical Studies

No studies were identified that were specifically designed to evaluate the safety and tolerability of GABA. However, GABA has been studied extensively in clinical studies for different applications including treatment of insomnia, high blood pressure, and stress and as an ergogenic substance to increase growth hormone (GH). We examined some of these studies that contained information relevant to safety as part of the USP comprehensive review of GABA. 

Numerous studies have examined the effect of supplementation with GABA in dietary supplements or in functional foods (e.g., studies on GABA powder in capsules, added to rice, or produced naturally in fermented milk or fermented soy). Most of these studies examined the effect of GABA on mild hypertension [50,51,52,53,54,55,56,57,58,59,60,61,62,63,64,65]. In total, 16 studies were identified that investigated the effect of orally administered GABA as a supplement or in complex matrices (such as fermented milk and soy sauce) on high blood pressure, for relieving stress, or for enhancing sleep. GABA was tested in doses ranging from 0.25 mg to 18,000 mg/day for periods of 4–12 weeks. Some studies also investigated the effect of a single intake of GABA [52,65,66,67,68,69,70,71,72,73]. All the studies reported that GABA was not associated with any serious adverse effects, although GABA was associated with a transient, moderate drop (<10% change) in blood pressure (BP). BP returned to baseline values a few days after the participants stopped taking GABA, as described below. 

### 6.2. Effects of GABA on Blood Pressure

A double blind, placebo-controlled, parallel-design study examined the tolerability of GABA-enriched food in 177 hypertensive men and women who were not undergoing treatment for the hypertension [74]. Participants were randomly allocated to receive 8 mL of one of the following: low-salt soy sauce containing 120 mg of GABA, low-salt soy sauce (control), or regular soy sauce (control), daily each for 12 weeks. At the end of the study, the average systolic blood pressure (SBP) was lower by 4.6 mm Hg (*p* < 0.05) in the GABA-treated group compared to the control groups. Diastolic blood pressure (DBP) was not changed in all treatment groups, and average BP was above normal in all groups at the end of the study with no difference between groups. Additionally, significant changes were observed in lactate dehydrogenase (LDH), albumin, urea nitrogen, and calcium, although the changes were below a 10% difference and were considered within the historical reference range and thus clinically irrelevant. Incidence of adverse effects were similar between groups and not considered due to the GABA [74].

A randomized, placebo-controlled, signal-blind trial investigated the effects of a fermented milk product containing GABA (FMG) on blood pressure in 39 patients with mild hypertension [62]. Patients received 100 mL of FGM daily containing 10–12 mg of GABA for 12 weeks or 100 mL of placebo for 12 weeks followed by 2 weeks of no intake. The placebo was a mixture of L-Lactic acid and sweeteners added to skim milk to mimic the taste of FGM. A significant decrease in BP was observed by weeks 2–4 in the FGM groups, and BP remained lowered throughout the 12-week period. In the FGM-treated group, BP decreased by 17.4 mm Hg for SBP and 7.2 mm Hg for DBP and was significantly different from baseline (*p* < 0.01) and from the placebo group (*p* < 0.01). Measurements of heart rate, body weight, hematologic and blood chemistry variables, glucosuria, and proteinuria were within the normal historical range in all treatment groups. Although none of the patients reported any side effects, four FGM group patients dropped out of the study—two for personal reasons, one who used an antihypertensive drug, and one who developed mild stomach ulcers not considered due to the intake of FGM [62].

In a placebo-controlled, double-blind study, the effect of fermented milk containing GABA (prepared using *L*. *casei* and *L*. *lactis)* was studied in 86 healthy subjects with mild or moderate hypertension [63]. Participants received either 100 mL of plain skim powdered milk (formulated to contain a similar amount of lactic acid as the milk containing GABA) or fermented milk providing 10 mg of GABA daily (0.15 mg/kg body weight/day) every morning for 12 weeks. Participants were evaluated clinically and interviewed, and the following variables were determined at intervals: weight, body mass index (BMI), blood pressure, heart rate, urine indices, and standard blood clinical chemistry and hematology. All parameters were unremarkable except for BP, which remained slightly elevated in both groups; however, the average systolic and diastolic blood pressures were slightly but significantly higher in the GABA group by ~5% (*p* < 0.01) compared to the controls. No GABA-related adverse effects were reported [63]. 

In a second randomized, placebo-controlled, double-blind trial, Kajimoto et al. [64] assessed the effects of GABA in 108 healthy men and women with high-normal blood pressure (42 men and 66 women). Average age was 46.4 ± 1.7 and 47.1 ± 1.7 years and weight was 62 ± 1.4 and 61.3 ± 1.3 kg in the test and placebo groups, respectively [64]. Participants received 100 mL of either placebo or the same fermented milk as described above by Kajimoto et al. [63], providing approximately 12.3 mg GABA/day every morning during the 12 weeks in the supplementation phase of the study. Participants were assessed for changes in weight and BMI, and standard clinical chemistry, hematology, urinalysis, and BP were determined one week before beginning the study, at the end of the study, and at intervals during the study. BP measurements in the GABA-supplementation group were above normal throughout the study in both groups, but the average SBP and DBP decreased by ~7% (*p* < 0.05). All other measurements were unremarkable and similar in the placebo and GABA-treated groups, and no GABA-related side effects were observed during the study [64]. 

In a third study, Kajimoto et al. [64] investigated the supplemental use of GABA in 88 hypertensive yet otherwise healthy subjects (31 men and 57 women; 53.8 ± 8.5 and 54.7 ± 8.6 years and weighing 59.7 ± 10.1 and 58.8 ± 9.2 kg for test and placebo groups, respectively) using a randomized, double-blind, placebo-controlled, parallel group study design [64]. Following a 2-week observation period, subjects were randomly assigned to receive either four placebo or four GABA tablets (Otsuka Pharmaceutical Co., Tokyo, Japan; 20 mg GABA/tablet) daily for a period of 12 weeks, resulting in either 0 or 80 mg GABA/day. A 4-week post study observation period was also included in the trial. BP and body weight were measured, and blood (clinical chemistry, including plasma GABA levels, and hematology) and urine tests were performed [64]. Similar to earlier studies, plasma GABA levels were not significantly increased relative to controls after 12 weeks of consuming GABA at a dose of 80 mg/day. A significant but transient change in BP was observed (−5%; *p* < 0.01) in subjects receiving GABA compared to those in the placebo group. Although some clinical chemistry values were decreased in the GABA group, all values were within historic control ranges and were considered to be clinically irrelevant. Urinalysis findings were considered unremarkable. No GABA-specific adverse effects were reported by any of the subjects, and none of the reported symptoms (cold, headache, diarrhea, loose stools, hand-foot-mouth disease in one patient, itching, or rash) were deemed by the study investigator(s) to be related to GABA administration.

Another study investigated tolerability of GABA supplementation in mildly hypertensive but otherwise healthy adults [61]. The authors first established an optimum dose in mildly hypertensive subjects (SBP between 130 and 180 mm Hg) who were randomized to receive oral doses of GABA at 0 (placebo), 20, 40, or 80 mg/day for 4 weeks. Hematologic, clinical chemistry, and urinalysis findings were unremarkable after 4 weeks of GABA supplementation, and measurements of standard safety-related parameters were unremarkable at all doses. An intake of 80 mg/day of GABA was associated with a significant reduction of the BP in adults with mild hypertension, and no adverse effects were reported. A subsequent study evaluated long-term effects of GABA at 80 mg daily versus placebo in mildly hypertensive subjects for 8 weeks. At the end of the 8-week study, SBP and DBP were on average 5% lower (*p* < 0.05) in all the subjects who received 80 mg/day of GABA compared to participants in the placebo group whose BP levels remained above normal [61]. 

### 6.3. Effects of GABA on Growth Hormone Levels

GABA has been reported to increase serum GH levels and has been considered an ergogenic aid. Many sports supplements include GABA as an ingredient.

Three studies by Cavagnini et al. published in 1980 and 1982 evaluated the effect of high-level supplementation with oral GABA as a single 5-g dose on GH levels in nine female subjects [67]. In one study, the plasma level of GABA was significantly (*p* < 0.0001) elevated compared to the level in subjects who were administered a GABA antagonist prior to taking GABA. Prolactin level was unaffected in the group administered only GABA. The authors reported that some of the participants who took a single high dose of GABA experienced a slight burning sensation in the throat that was accompanied by breathlessness in some cases. This was transient and resolved without treatment, although the duration of the effect is not indicated [67]. The second study by Cavagnini et al. administered a single 5-g dose of GABA dissolved in 150 mL tap water to 19 participants (2 male and 17 female) and 150 mL of tap water as placebo to 18 participants (1 male and 17 female). For an insulin tolerance test, eight female participants were administered 18 g of GABA in four divided doses daily for 4 days, with the last dose given an hour before a post-study insulin tolerance test [66]. The 5-g dose of GABA was associated with significant increases in GH levels (*p* < 0.0001) to above 5 ng/mL. Again, some participants reported a burning sensation in the throat immediately after ingesting GABA, and in some cases, this was accompanied by breathlessness, which returned to normal shortly after. Some participants also reported lethargy and weakness in the legs. In the group that received 18 g of GABA for 4 days, there was a significant (*p* < 0.01) blunting of overall GH release, but prolactin level was significantly increased in response to insulin hypoglycemia test. In all the other groups, there was no effect on prolactin levels, pulse rate, BP, or baseline blood glucose concentration [66]. 

In a third study, Cavagnini et al. [50] investigated the potential effect of GABA on pancreatic function. Twelve healthy individuals (three men, nine women) were divided into three groups, and each group received three doses of either placebo or 5 g or 10 g of GABA dissolved in water. The three doses were given on separate days, 2–3 days apart. After GABA administration, plasma levels of immunoreactive insulin (IRI), C peptide (CP), immunoreactive glucagon (IRG), and glucose were determined. There were significant (*p* < 0.001) increases in IRI, CP, and IRG in the group that received 5 g or 10 g of GABA, although the effect was transient and returned to baseline within 180 minutes. There were no changes in blood glucose levels following GABA consumption, and all participants in the GABA group completed the study. There were no reports on side effects following administration of GABA [50]. Another study of GABA effects in resistance-trained men showed that an intake of 3 g was associated with approximately 400% higher concentrations of both immunoactive and immunofunctional GH. GABA intake combined with resistance exercise was associated with 200% higher immunoactive and 175% higher immunofunctional GH levels compared to resistance exercise alone. No adverse effects were mentioned for this study [58]. 

### 6.4. Effect of GABA on Sleep and Stress

A prospective, randomized, double-blind, and placebo-controlled trial evaluated the effects of GABA extracted from unpolished rice germ for improving sleep quality in 40 patients with insomnia [59]. Participants took 300 mg GABA or 300 mg maltodextrin (placebo) daily for 4 weeks. The 300-mg dose was selected based on an earlier study in which patients reported their subjective insomnia improved more with 300 mg of GABA compared to 150 mg [75]. Participants filled out a sleep questionnaire and underwent polysomnography before and after the study. After 4 weeks of GABA supplementation, sleep latency was significantly decreased (*p* = 0.001), and sleep efficacy was significantly increased (*p* = 0.018) compared to placebo. Three out of 10 subjects given GABA had either abdominal discomfort, headache, or drowsiness, which were classified as mild to moderate. The authors concluded that GABA supplementation may improve sleep quality without serious adverse effects [59]. 

In a study that evaluated the effects of GABA on heart rate variability and stress, there was no mention of adverse or untoward effects in healthy volunteers given a GABA-enriched oolong tea (2.01 mg of GABA per 200 mL tea, as analyzed using HPLC). Consumption of GABA-enriched tea was associated with an improved heart-rate variability and a significant decrease in the immediate stress score compared to volunteers given regular tea (not enriched with GABA). Adverse events were not reported as being monitored in this study, and the authors recommended that safety and tolerability of GABA intakes should be investigated in future studies [68].

### 6.5. Other Studies

One placebo-controlled study evaluated the effects of *Laminaria japonica* on short-term memory and physical fitness, and no side effects were mentioned in elderly volunteers given 1.5 g/day of fermented *Laminaria japonica* for 6 weeks. The mean content of GABA in fermented *Laminaria japonica* was 54.5 mg/g, providing an intake of approximately 81.75 mg/day of GABA. Although adverse reactions were monitored, there was no mention of any adverse effects occurring during the study [76]. 

A clinical trial proposed to evaluate the safety of GABA supplementation in patients with longstanding type I diabetes mellitus is listed on the NIH clinicaltrials.gov website (Identifier: NCT03635437). The goal of the proposed trial is to find a reasonably safe and tolerable treatment in type I diabetic adults that will restore some endogenous insulin secretion, improve quality of life, and reduce the risk of both short- and long-term complications. The intake levels proposed in the trial are 200 or 600 mg/day of GABA for 6 months and 600 mg/day of GABA combined with 0.5 mg/day of oral alprazolam for 3 months followed by 600 mg/day of GABA alone for another 3 months. The trial is currently in the recruiting phase [77]. 

The Cochrane library produced no relevant article for the search terms “GABA” or “Gamma-Aminobutyric Acid” [78]. Articles retrieved were about pregabalin and gabapentin.

### 6.6. Adverse Events Associated with GABA Intake

The clinical studies reviewed above did not associate any serious adverse events with GABA intake. Some adverse effects that were reported following intake of GABA included abdominal discomfort, headache, drowsiness, and transient burning sensation in the throat; these effects were classified as mild to moderate [50,59,61,67,75]. No case reports of AEs associated with GABA were found in the literature. Searches were conducted in various government reporting portals as described below.

A search of the publicly available FDA (CAERS) database using the terms “GABA” and “gamma-aminobutyric acid” yielded 292 AERs and 156 AERs, respectively, which included duplicates. Of these, 151 (97%) were associated with multi-ingredient DS, and only 3% were associated with single-ingredient products. The adverse effects reported in the single-ingredient AERs were all different. Causality assignment was not possible due to the limited information available.

A search of the Canada Vigilance Program [79] publicly accessible database yielded one spontaneous serious AE in a 23-year-old male who had been taking 500 mg of GABA orally for 247 days. The patient presented with anxiety, balance disorder, depression, hyperhidrosis, insomnia, mood swings, paresthesia, self-injurious ideation, abnormal thinking, and withdrawal syndrome. The patient was concurrently taking 5-hydroxytryptophan, which was also considered suspect, and B-complex 100 timed-release tab and Hair Force (a multi-ingredient product); all were considered concomitant medications. Because the patient was using other suspect products, the role of GABA, if any, in the adverse effects is uncertain. Searches in other public databases including the Medicines and Healthcare Products Regulatory Agency (MHRA, London, United Kingdom) and Australia Therapeutic Goods Administration (TGA) [80] yielded no reports.

## 7. Animal Toxicology and In Vitro Studies

### 7.1. Acute Toxicity Studies

A single-dose toxicity study was conducted by Japan Food Research Laboratories (JFRL) [81] in 4-week-old male and female rats (10/sex/group). The test material contained 80% pure GABA (denoted GABA-80) and was diluted 4-fold with dextrose to obtain 20% GABA (denoted GABA-20). Rats were administered GABA-20 dissolved as a single dose in water to provide 5000 mg/kg body weight of GABA-20, which would correspond to an acute GABA exposure of 1000 mg/kg body weight. The control, non-treated group received water. Rats were monitored for clinical signs and mortality over 14 days. No clinical signs or deaths occurred, and no differences in body weight were observed throughout the study in the treated rats compared to controls. Because there was no evidence of morbidity and no deaths, the LD_50_ for GABA-20 was considered to be >5000 mg/kg body weight (or approximately >1000 mg GABA/kg body weight). Previous studies had reported higher oral LD_50_ values of 12,000 mg GABA/kg in mice [82]. 

### 7.2. Sub-Chronic Toxicity Studies

A 28-day toxicity study was conducted by the Japan Scientific Food Association in male and female Wistar rats (Hayami et al., 2005). The test article was GABA-20 containing 20% GABA, identical to material that was used in the acute toxicity study of JFRL (2002) [81]. Rats (32 days old, 20 of each sex per group) were fed either plain chow or chow supplemented with 1% GABA-20 (providing approximately 1000 mg GABA-20/kg body weight/day corresponding to approximately 200 mg GABA/kg body weight/day). Rats were monitored for clinical signs, body weight gain, and food consumption. At the end of the study, hematologic and biochemical tests, standard gross pathology, and measurement of organ weights were performed. Histopathological examination was performed on brain, heart, liver, kidney, testes, and ovaries. There were no significant differences between treatment and control groups in weight gain and average food consumption. No signs of morbidity were observed, and no mortality was reported. There were no remarkable changes in histopathology, hematology, or biochemical parameters in treated animals relative to controls, and necropsy showed no abnormalities. The only significant difference in organ weights was the absolute and relative testis weights, which were increased compared to baseline in both control and treated rats by 7% and 6% (*p* < 0.05), respectively. However, the authors considered this increase to be toxicologically insignificant as they were within the range of historical data [83]. 

A 90-day toxicity study administered GABA by oral gavage at doses of 500, 1250, and 2500 mg/kg body weight to groups of 10 male and 10 female Sprague–Dawley rats for 13 weeks. The data collected included clinical parameters, body weight, food consumption, ophthalmology, hematology, blood chemistry, and urinalysis; full necropsy was done including determination of organ weights. GABA was well tolerated, and no deaths occurred that could be attributed to the test material. Side effects observed included mild diarrhea in five males and one female among rats administered 2500 mg/kg/day, and there was temporary salivation immediately after administration in eight males and seven females in the high-dose group, which resolved without intervention. One male in the 1250 mg/kg/day group was found dead on day 88; however, the death was considered random and unrelated to GABA administration since no histopathological changes were observed at necropsy. A few males showed significant changes in body weight gain; however, there was no dose-response relationship, and thus the changes were attributed to food intake. Organ weights were similar between groups, and the results of the histopathological examinations were unremarkable. The authors concluded that oral GABA administration of up to 2500 mg/kg was well tolerated. They noted that the minor observed changes in some males in clinical signs, hematology, clinical chemistry, and histopathology were within the historical ranges and were not dose-dependent and thus were not considered toxicologically significant. No changes were observed in females. There were no other notable findings that could be attributed to the administration of GABA to rats at doses of up to 2500 mg/kg body weight/day for 13 weeks, including no significant observations in clinical signs, mortality, decreased body weights, hematology, blood chemistry, and organ weights [84]. 

A review conducted by the EPA indicated that studies in the literature involving prolonged chronic administration of large doses of GABA to rats and dogs (up to 1 g/kg/day) reported no signs of toxicity or untoward effects ((Federal Register Volume 62, Number 209 (Wednesday, 29 October 1997)).

One study found that oral dosing with 25, 50, or 75 mg/kg/day of GABA for 14 days ameliorated fluoride-induced hypothyroidism in male Kumning mice (hypothyroidic mice) [85]. Long-term thyroid hormone therapy is often associated with side effects on the heart in patients with hypothyroidism, but oral GABA treatment was without side effects on the myocardium of hypothyroidic mice compared to negative control hypothyroidic mice (without GABA treatment) and positive control hypothyroidic mice (treated with unspecified thyroid medication for 14 days). Both negative and positive control groups exhibited histopathology findings in the myocardium consisting of irregular arrangement and rupture of myocardial fibers along with nuclear pyknosis and swelling, whereas hypothyroidic mice treated with GABA showed normally arranged myocardial fibers with decreased swelling and pyknosis of nuclei [85].

### 7.3. Genotoxicity

A DNA repair test conducted in *Bacillus subtilis* strains H17 (Rec+) and M45 (Rec−), examined the mutagenicity of two fermentation products containing GABA at concentrations greater than 10 and 25 mg of GABA per 100 mL. A dose-range-finding study and main studies were carried out with or without metabolic activation using S9. Both products did not show any DNA damage in all studies in the presence or absence of the S9 mix. A negative control (saline) was included and did not show DNA damage, while the positive control (2-aminoantracene (2-AA) and 2-(2-furyl)-3-(5-nitro-2-furyl)-acrylamide (AF-2)), as expected, showed mutagenic ability [86]. 

### 7.4. Reproductive Effects of GABA

No reproductive or developmental toxicity studies were found for GABA. However, in vitro studies showed that GABA may be involved in the fertilization process by enhancing capacitation of sperms, modulating placenta trophoblasts, and stimulating androgen production. An in vitro study of the effects of GABA on human sperm motility and hyperactivation determined that GABA increased sperm kinematic parameters and hyperactivation, similar to the effects of progesterone on human sperm. The effect of GABA and progesterone together was no different than their effects separately, and their effects were blocked by bicuculline, a GABA_A_ receptor antagonist [87]. 

In one experiment where semen from Australian Merino rams was incubated with GABA at various concentrations ranging from 1 to 20 µM, a marginal capacitation of sperms was observed at 1 µM, which increased to a maximum capacitation at 20 µM. Additional experiments showed that incubation of ram semen with 1 µM of GABA and the steroid allopregnanolone, an allosteric modifier of the GABA_A_ receptor, resulted in a significant increase in sperm capacitation similar to that observed for 20 µM, indicating that the GABA capacitation effect is mediated via a GABA_A_ receptor-mediated mechanism [88]. 

In other studies, GABA was shown to stimulate acrosome reactions in pre-capacitated human spermatozoa in a concentration-dependent manner. The effect was dependent on the availability of extracellular Ca21 because the inclusion of EGTA or La31, a Ca21 channel agonist, prevented GABA-induced acrosome reactions [89]. Similarly, in mouse sperms, GABA promoted acrosome reaction, and the effect was suppressed by GABA_A_ receptor agonist. Furthermore, GABA facilitated the tyrosine phosphorylation of sperm proteins, an index of sperm capacitation [90]. The GABA effects of capacitating sperm and stimulating acrosome reactions are significant, as these activities facilitate fertilization. Mammalian spermatozoa undergo an acrosome reaction in response to oocyte agonist(s), an essential process that results in the release of enzymes necessary for sperms to penetrate the egg vestment, allowing the sperms to fuse with the oocyte’s plasma membrane after penetration of the zona pellucida [91]. Other experiments have shown that GABA regulated the biosynthesis of hCG in human first trimester placenta acting via the GABA_A_ like receptors [92]. 

Finally, an in vitro study using rat testes obtained at different stages of maturation suggested that GABA plays a physiological role in the regulation of rat testicular androgen production, although this may be dependent of the stage of sexual maturation. GABA at a concentration of 10^−6^ modified the basal and hCG-stimulated androgen production in testes obtained from adult (60 days), pubertal (45 days), and pre-pubertal (31 days) rats. However, pre-pubertal testes exposed to the same concentration of GABA (10^−6^) showed much lower stimulatory effects on hCG and a significant increase in androstanediol production [93]. 

## 8. Pharmacokinetics of GABA

Because of its importance as a neurotransmitter, GABA’s metabolism has been well characterized in humans and animals [39,94,95,96]. The liver is considered to be the primary metabolic site for extra-cerebral GABA, and rats display a large capacity for GABA uptake by this organ [97,98]. Some animal studies have shown that oral dosing does not increase GABA plasma levels significantly. In one study in rats, following a one-time oral administration of 500 mg GABA per kg body weight, the plasma level of GABA remained at approximately 1.6 µM/mL (similar to baseline) when measured immediately after oral administration and 120 minutes later. Other studies reported that following intraperitoneal administration of the 500 mg GABA per kg body weight, plasma levels rose to approximately 400,000 µM/mL and progressively decreased to 1.2 µM/mL after 120 minutes. This could be an indication that in rodents, the absorption and/or bioavailability of orally administered GABA is very low. GABA was shown to undergo rapid clearance with a half-life of approximately 20 minutes in rats, rabbits, and cats following oral administration [97]. Other studies have shown that following systemic administration in the rat and mouse, GABA is distributed primarily to the liver, kidneys, and muscle. In the mouse, significant amounts of GABA were detected in the urinary bladder, gastrointestinal wall, pituitary gland, and cartilage of the spine, ribs, and trachea [97,98]. No evidence of GABA bioaccumulation or organ-specific retention was reported in any of the reviewed studies. 

GABA absorption by the intestine is mediated via carrier proteins normally involved in nutrient absorption and appears to involve H^+^/zwitterionic GABA cotransport [99]. Studies using rat intestine suggest that GABA shares a transporter with β-alanine [100]. This may explain why very little GABA is bioavailable when ingested orally. Catabolism of GABA occurs exclusively via GABA transaminase, during which GABA is converted to the metabolite succinate semi-aldehyde. This can then be reduced to gamma-hydroxybutyrate or oxidized to succinate and eventually converted to CO2 and water via the citric acid cycle (Krebs cycle). Thus, GABA is essentially utilized as an energy source in the body and is metabolized to innocuous compounds. 

In one open-label, three-period clinical study where participants ingested 2 g of GABA once and 2 g of GABA three times daily for 7 days, with 7-day wash-out between periods, GABA was rapidly absorbed (T_max_: 0.5~1 h) with a half-life of 5 h. No accumulation of GABA was observed following repeat oral administrations [101]. Participants who received repeated doses of GABA showed higher incidences of minor adverse events, including sore throat, throat burning, a skin burning sensation, headache, and dizziness. However, there were no clinically relevant changes in all participants in vital signs, EGCG parameter, physical examination, hematology, biochemistry, and urinalysis. No serious adverse events were observed.

Because GABA is a key neurotransmitter in the CNS, it is important to understand how much of the administered amount may traverse the blood-brain barrier (BBB) when it is ingested orally as a dietary supplement. So far, evidence indicates that very low amounts of GABA cross from the plasma into the brain through the BBB even when GABA is exogenously administered orally or intravenously [97,98,102,103,104,105,106,107,108].

In one rat study, increasing the administration of GABA (intraperitoneally) by 1250-fold resulted in only a 30-fold increase in the levels of GABA in the cerebrospinal fluid (CSF). In other studies, an increase in plasma concentrations was not associated with an increase in GABA permeation rates across the BBB of rats [105]. The absence of a dose-response relationship between the administered dose of GABA and GABA levels in the brain may be partially explained by findings of a study by Kakee et al. (2001) showing that the GABA efflux rate through the BBB of rats exceeded influx by approximately 16-fold [109] and other studies indicating that GABA transaminase can rapidly degrade very large amounts of GABA when administered intracerebrally, thereby decreasing the plasma levels [102]. There appears to be an internal regulating mechanism that maintains the balance of GABA plasma levels irrespective of whether external GABA is administered. 

## 9. Potential Interactions of GABA with Drugs 

Some clinical studies [57,110,111] and animal experiments [110,112,113,114] have shown that ingestion of GABA may cause a drop in blood pressure, and thus it is conceivable that GABA may interact with antihypertensive medicines, such as propranolol, metoprolol, etc. GABA has been reported to cause a BP decrease of approximately 10%, but the effect was transient, with BP returning to baseline within a few days after stopping the GABA [74]. 

Evidence from at least one clinical study shows that the bioavailability of GABA in the brain is improved significantly when taken concurrently with phosphatidylserine, and thus GABA may interact with medicines used to treat epilepsy [115]. It has been suggested that GABA supplementation could affect the enteric nervous system and possibly stimulate the endogenous production of GABA across the BBB [116,117]. 

The oral administration of a mixture of GABA and L-theanine (100/20 mg/kg) to ICR mice was found to decrease sleep latency and prolong sleep duration compared to GABA or L- theanine treatment alone following an intraperitoneal injection with sodium pentobarbital. The authors concluded that GABA and L-theanine had a synergistic effect on the sleep behavior of mice [118].

## 10. Safety of GABA as Dietary Ingredient

In clinical studies, GABA taken orally at up to 120 mg/day for 12 weeks was not associated with adverse effects [74]. At much higher doses of 5 g/day and 10 g/day, the only mild side effect noted was a slight burning sensation in the throat, which disappeared after a few minutes [66]. Interestingly, another study also by Cavagnini et al. [66,67] that administered 18 g/day of pure GABA did not observe any adverse effects [67]. Another clinical study that administered up to 6 g daily (taken in divided doses of 2 g three times daily) for 7 days observed no serious adverse effects [101]. 

Labels of GABA-containing products found in the DSLD recommend intake amounts of up to 3 g per day (range of 45 mg to 3000 mg per day), although a majority of products (>70% of the 38 sampled) recommended 600–750 mg/day in divided doses. The recommended intake amounts are well below intake amounts associated with mild side effects in the Cavagnini et al. studies [50] that administered 5 g or 10 g per day and 18 g per day for 4 days with no serious adverse effect observed [66,67]. 

## 11. Concluding Remarks

This review was conducted as part of a dietary supplements admission evaluation that is performed for dietary ingredients prior to admission into the USP monograph-development process. The intent of the review is to determine whether an ingredient is associated with any serious risk(s) to health that would preclude its admission for monograph development. This report is a result of a review of available preclinical and clinical safety information and does not include a review of mechanistic data, as this was considered to be outside the scope of determining whether GABA is associated with serious risk to health. Where applicable and necessary, we have mentioned the mechanism of action of GABA. For example, we note that the GABA sperm capacitation effect is mediated via a GABA_A_ receptor-mediated mechanism [88]. We also provided information that there appears to be an in vivo internal regulating mechanism that maintains the balance of GABA plasma levels irrespective of whether external GABA is administered [102]. The review found no clinical studies specifically designed to study the safety of GABA in healthy individuals. Three short-duration clinical studies (a one-time 5-g dose, 18 g for 4 days, and 5 g or 10 g daily for 5 days) administered up to 18 g of GABA, and in all three studies, only a few participants reported a slight burning sensation in the throat immediately after taking GABA which ceased shortly thereafter. In some cases, the burning sensation was accompanied by brief shortness of breath. Additionally, intake of 5 or 10 g daily for 4–5 days resulted in increases in immunoreactive insulin and glucagon, although no change in blood glucose level was observed [50]. Taken together, no serious side effects were associated with the administration of pure GABA at doses of up to 18 g daily for 4 days.

Clinical data from 16 studies that examined the effect of GABA in different matrices (fermented milk or soy) on mild hypertension, insomnia, and stress and as an ergogenic substance at doses ranging from 10 mg to 12 mg/day for up to 12 weeks or 120 mg of GABA/day for 12 weeks reported that ingestion of GABA was not associated with any serious adverse effects [50,61,62,63,64,66,67,74,119]. Some participants experienced a moderate drop (≤10% change) in BP, which returned to baseline level a few days after the participants stopped taking the product containing GABA. No case reports associated with the ingestion of GABA were found. The current review found one spontaneous, serious adverse event report in the CVP database involving a 23-year-old male who had been taking 500 mg of GABA orally for 247 days. Because the patient was using other suspect products, the role of GABA, if any, in the adverse effects is uncertain. FDA MedWatch yielded 156 AERs, of which 151 (97%) were associated with multi-ingredient DS, and only 3% were associated with single-ingredient products. The adverse effects reported in the single-ingredient AERs were all different, and thus causality could not be determined due to the limited information available.

No studies of the effects of GABA in pregnancy and lactation were found. A number of in vitro studies of the effects of GABA on human sperm motility and hyperactivation determined that GABA was associated with sperm kinematic parameters and hyperactivation, similar to the effects of progesterone on human sperm [87]. Because of its effects on neurotransmitters and the endocrine system (increases growth hormone and prolactin) and the absence of data supporting its use during pregnancy and lactation, caution may be advisable in the use of GABA during pregnancy or lactation.

Acute toxicity studies of GABA in Sprague–Dawley rats at doses of 1000 mg/kg body weight determined a LD50 >1000 mg GABA/kg body weight [82]. A 28-day study and a 90-day study in rats determined a no-observed-adverse-effect level (NOAEL) at 5 mg GABA/kg body weight per day, the highest dose tested [120]. In a 90-day study, urine volume was significantly increased in males only at the highest concentration tested, 5 mg GABA/kg body weight. However, this was not associated with other adverse effects on urinary tract function, and thus the authors considered the effect not significant. A decrease in hemoglobin levels was seen in some GABA-treated females, but this was not accompanied by anemia and was not seen in males; thus, the authors concluded that this change was not significant.

Except for one clinical study that showed increased bioavailability of GABA in the brain when taken concurrently with phosphatidylserine, no other studies were identified on interaction of GABA with medicines or supplements. However, some clinical and animal research shows that GABA may decrease blood pressure in hypertensive subjects, so it is conceivable that concurrent use of GABA with medicines for hypertension might increase the risk of hypotension.

Based on this review, the USP Dietary Supplements Admission Committee has admitted GABA for USP monograph development. The proposed monograph is under development and planned for publication in the *USP Pharmacopeial Forum* in 2022.

## Figures and Tables

**Figure 1 nutrients-13-02742-f001:**
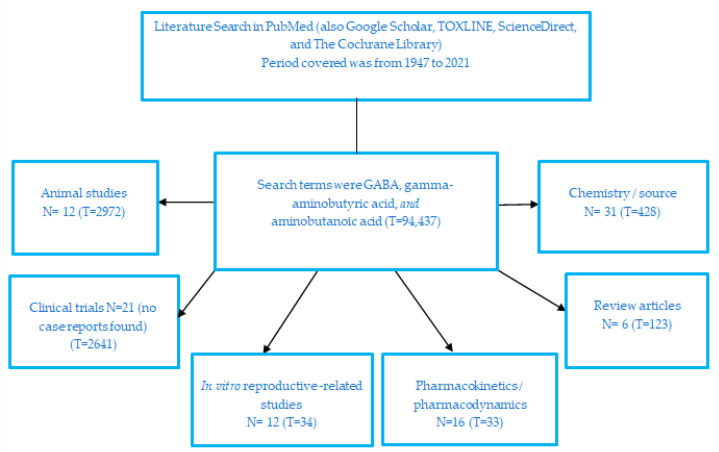
Literature search strategy showing the combination of search terms used and the number of articles included in the manuscript and the total number of articles retrieved by each search for each outcome. Most publications were concerning GABA as a neurotransmitter, its metabolism, and its derivatives and prodrugs such as pregabalin, gabapentin, and articles describing GABA function as an innate metabolite in the human body. T, total articles retrieved; N, articles included in the review.

**Figure 2 nutrients-13-02742-f002:**
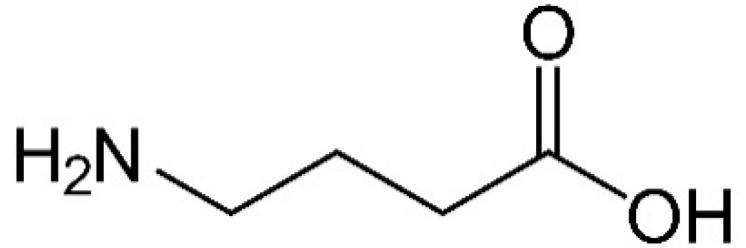
*Gamma*-aminobutyric acid (GABA, 4-aminobutyric acid).

**Figure 3 nutrients-13-02742-f003:**
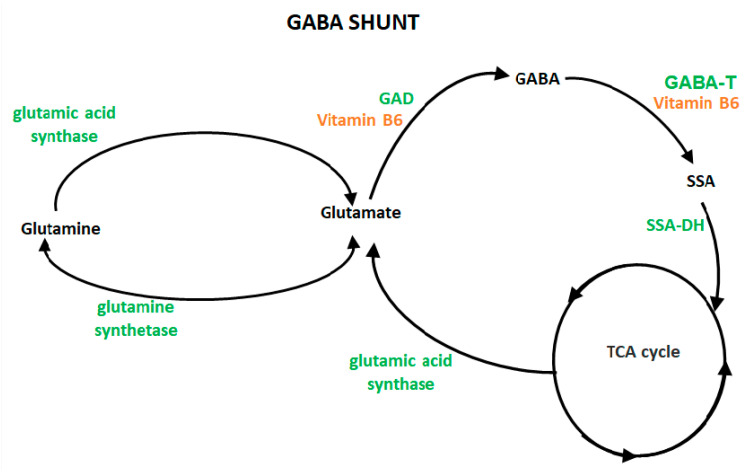
In the GABA shunt, GABA is synthesized from glutamate in a process catalyzed by GAD. GABA is metabolized by GABA-T into succinate semi-aldehyde, which is then reduced to gamma-hydroxybutyrate or oxidized to succinate and eventually converted to CO_2_ and water via the TCA cycle. GAD, glutamatic acid decarboxylase; GABA-T, GABA transaminase; SSA, succinic semialdehyde; SSA-DH, succinic semialdehyde dehydrogenase; TCA cycle, tricarboxylic acid cycle. Green, enzymes; Orange, cofactor.

## Data Availability

Not Applicable.

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
