# Peer review of "United States Pharmacopeia (USP) Safety Review of *Gamma*-Aminobutyric Acid (GABA)"

_nutrients, 2021, doi:10.3390/nu13082742_

Round 1

Reviewer 1 Report

The present manuscript by Hellen A Oketch-Rabah et al, is a comprehensive review of the current medical literature assessing the effects of gamma-aminobutyric acid (GABA) on health.

Based on the current trend of marketing GABA as a dietary supplement it was necessary to investigative the most recent publications to understand if the recommendation by commercial companies will not affect consumers health.

Studies had shown that GABA has been used for a variety of clinical applications to treat insomnia, high blood pressure, stress-anxiety and as an ergogenic substance to increase growth hormone.

The review is centered on data obtained from publications studying the clinical outcomes after GABA exposure in humans or laboratory rodents using different routes of administration, dosages and acute or chronic treatment.

They conclude that with the present knowledge GABA has no toxic effects given orally, has a short half-life, does not accumulate in peripheral tissues and does not crosses the blood brain barrier. More studies are needed to understand its effects in pregnancy and lactation.

Although the data present is interesting, these were observational and descriptive studies studies that were not followed by any mechanistical data (main weakness). Also, the same comment is raised for the background and introduction. It is possible that for purpose of the USP that is not necessary.

Therefore, I recommend this manuscript to be accepted with the addition and acknowledgement of their weaknesses. 

Reviewer 2 Report

The manuscript reports a safety evaluation intended to determine whether GABA is associated with any serious risks to health that would preclude its admission for monograph development. A comprehensive literature search about the safety, toxicology and pharmacokinetics of GABA was performed, and the authors found that 1) no serious adverse events associated with GABA at intakes up to 18 g/d for 4 days and in longer studies at intakes of 120 mg/d for 12 weeks was identified; 2) GABA was associated with a transient and moderate drop in blood pressure; 3) No studies of the effects of GABA in pregnancy and lactation were identified; 4) Acute toxicity studies determined an LD50 >1,000 mg GABA/kg body weight in Sprague-Dawley rats, and no signs of toxicity was found in chronic administration of GABA to rats and dogs at doses up to 1 g/kg/day. In general, this manuscript is well written, and is thus recommended for publication. However, there are still a few minor issues that need to be addressed:

  1. Page 2, line 60, define ‘DS’.
  2. Page 3, line 113-114, change the scale of GABA in mg/g to nmol/g.
  3. Fig 3, correct the word ‘svnthetase’ in the figure.
  4. Page 4, line 153, define NLT/NMT
  5. Page 4, line 154 and 158, define IR, HPLC, GC
  6. Page 5, line 218, change the font to match with others
  7. Page 14, line 670-671, the font looks different.
